# Strawberry *FaWRKY25* Transcription Factor Negatively Regulated the Resistance of Strawberry Fruits to *Botrytis cinerea*

**DOI:** 10.3390/genes12010056

**Published:** 2020-12-31

**Authors:** Sizhen Jia, Yuanhua Wang, Geng Zhang, Zhiming Yan, Qingsheng Cai

**Affiliations:** 1College of Life Sciences, Nanjing Agricultural University, Nanjing 210095, China; jiasizhen@jsafc.edu.cn; 2Department of Agronomy and Horticulture, Jiangsu Vocational College of Agriculture and Forestry, Jurong 212400, China; wangyuanhua0511@163.com (Y.W.); gengzhang@jsafc.edu.cn (G.Z.); yanzhim@jsafc.edu.cn (Z.Y.); 3Jiangsu Engineering and Technology Center for Modern Horticulture, Jurong 212400, China

**Keywords:** strawberry, *FaWRKY25* transcription factor, *Botrytis cinerea*

## Abstract

WRKY genes and jasmonic acid (JA) play a crucial role in plants’ responses against biotic and abiotic stress. However, the regulating mechanism of WRKY genes on strawberry fruits’ resistance against *Botrytis cinerea* is largely unknown, and few studies have been performed on their effect on the JA-mediated defense mechanism against *B. cinerea*. This study explored the effect of *FaWRKY25* on the JA-mediated strawberry resistance against *B. cinerea*. Results showed that the JA content decreased significantly as the fruits matured, whereas the *FaWRKY25* expression rose substantially, which led to heightened susceptibility to *B. cinerea* and in strawberries. External JA treatment significantly increased the JA content in strawberries and reduced the *FaWRKY25* expression, thereby enhancing the fruits’ resistance against *B. cinerea*. *FaWRKY25* overexpression significantly lowered the fruits’ resistance against *B. cinerea*, whereas *FaWRKY25* silencing significantly increased resistance. Moreover, *FaWRKY25* overexpression significantly lowered the JA content, whereas *FaWRKY25* silencing significantly increased it. *FaWRKY25* expression level substantially affects the expression levels of genes related to JA biosynthesis and metabolism, other members of the WRKY family, and defense genes. Accordingly, *FaWRKY25* plays a crucial role in regulating strawberries’ resistance against *B. cinerea* and may negatively regulate their JA-mediated resistance mechanism against *B. cinerea*.

## 1. Introduction

Strawberry (*Fragaria* × *ananassa* “Benihoppe”) is an important horticultural crop worldwide, being rich in vitamins and antioxidants and having excellent commercial value and health benefits [1] Strawberries can be affected by various pathogens during growth, such as fungi, bacteria, viruses, and nematodes; fungus is one of the most threatening types of pathogens to strawberries, and can infect all parts of strawberries, resulting in severely damage or death of plants [2]. *Botrytis cinerea* is an extremely destructive pathogen to strawberries; it causes *B. cinerea* on strawberries before and/or after harvest. *B. cinerea* has been regarded as the second largest fungal pathogen of strawberries due to its severe damage to the strawberry fruits [3]. It was reported that more than 80% of flower buds and fruits of strawberries may be lost as a result of infecting with *B. cinerea* under a humid environment without fungicides [4]. Recently, fungicides have been broadly applied to safeguard the production quality and quantity of strawberries and other major crops, but this incurs public concerns over food safety; therefore, sustainable alternatives to fungicides or disease-resistant germplasms must be developed [5]. Research has been conducted to clarify the effect of fungal pathogens on plants, but comprehensive findings on strawberries’ defensive molecular level and mechanism against *B. cinerea* are lacking [6,7,8]. Thus, identifying the defensive reactions of strawberries against this pathogen is fundamental to further understanding the potential defensive molecular mechanism of the fruits.

The resistance to invaders is frequently harmonized in plants through a complex defense molecular network fine-tuned by phytohormones such as salicylic acid (SA), jasmonic acid (JA), and ethylene (ET), which regulate the defensive response to efficiently face the different pathogens [9]. JA/ET signaling pathway is commonly activated in plants against necrotrophic pathogens or insects, or in response to wounding [10]. JA induces a different set of defense response genes and the production of a large variety of secondary metabolites such as alkaloids, phenolic compounds, and terpenes [11]. JA signaling molecules in plants make the plant responsive to various biotic and abiotic stresses and can control pollen maturation and accidental damage in *Arabidopsis* [12,13]. The defense-related phytohormone jasmonate has been described as an inductor of resistance during postharvest storage in many plant species. As a plant-signaling molecule, JA has been reported to enhance disease resistance in various fruits during postharvest storage [14,15]. Crosstalk among these signaling pathways has been well described in models [9], while remaining largely unknown or poorly understood in strawberry.

Transcription factors (TFs) are considered key regulators of gene expression, playing important roles within this complex defense molecular network and leading to plant immunity [16,17]. To date, many defense-related TFs have been identified in plants, including MYBs, the TGA/bZIP family protein, AP2/ERF-ET responsive element binding factors, NACs, the Whirly (WHY) family protein, and WRKYs [18,19,20].

WRKYs are the largest transcription regulator family in land plants. They can regulate the growth and development of plants as well as their response to biotic and abiotic stress [21]. WRKY genes have been confirmed to be related to plants’ reaction against *B. cinerea* and are critical for hormone-regulated disease resistance and defense signaling networks [22,23,24]. For example, most of the WRKY TFs in *Arabidopsis thaliana* react to the infection of pathogens [25] and positively or negatively regulate plant defense reactions [26]. The upregulated *AtWRKY70* expression in *A. thaliana* strengthens the plant’s resistance against *B. cinerea* and is a negative regulator of jasmonic acid (JA)-inducible genes [27]. The genes *AtWRKY18*, *AtWRKY40*, and *AtWRKY60* positively coregulate the plant’s resistance reaction against *B. cinerea* [28]. *AtWRKY33* controls the balance of multiple hormones and phytoprotection to regulate plants’ defense reaction against the pathogen [29]. WRKY TFs frequently exhibit dual activity in the plants’ defense mechanisms according to the types of pathogen. In the leaves of *A. thaliana*, *AtWRKY50* and *AtWRKY51* are positive regulators of salicylic acid-mediated signals and negative regulators of JA-mediated signals [30]. In the vegetative tissue of *A. thaliana*, *AtWRKY3* and *AtWRKY4*, which have similar structures, have been regarded as positive regulators of the plant’s resistance against necrotic pathogens such as *B. cinerea*; *AtWRKY4* also negatively regulates plants’ resistance against biotrophic pathogens such as *Pseudomonas syringae* [26]. Accordingly, WRKYs regulate plants’ defense reactions against *B. cinerea* in individual, collaborative, or complex forms. Although numerous studies have been conducted on members of the WRKY family in various crops, few have examined their functions in and regulations of the defense mechanism of strawberries, particularly that of the fruits.

Currently, 59 WRKY genes at different fruit growth stages of forest strawberries and 62 WRKY genes responding to biotic and abiotic stress have been identified [31,32]; moreover, 47 WRKY genes have been identified in octoploid strawberries [33]. *FaWRKY1* was the first strawberry WRKY identified as a mediator of defense response against *Colletotrichum acutatum* in cultivated strawberry [34,35]. *FaWRKY1* encodes an *AtWRKY75*-like transcription factor type IIc, which is upregulated after *Colletotrichum* infection and responds to defense-related hormones such as SA, JA, ABA, and wounding, with its expression being dependent on strawberry cultivar and tissue [35,36]. A recent study has shown that *FvWRKY42* is a positive regulator of the resistance of wild strawberries against powdery mildew [37]. These results confirmed the role of WRKY gene in strawberry disease defense response, but few have explored the genes specific to strawberry resistance against *B. cinerea*. Furthermore, a few studies have examined the functions of a single gene in the disease resistance of strawberries, and the gene functions have been verified through heterogeneous verification—the gene function verification method employed in *A. thaliana*. However, species differences may cause the regulatory function of a gene in pathogen biology to vary according to the plant species or tissues. Therefore, the aforementioned verification method cannot effectively identify the specific effects of genes in strawberries.

WRKY25 is a class I WRKY protein and features a highly conserved WRKY domain [38]. WRKY25 is a key TF in plants’ resistance against biotic and abiotic stress in *A. thaliana* and poplars. It is related to plants’ resistance against fungal pathogens and plays a critical role in their defense against them [39,40,41,42,43]. 

In order to further clarify the biological functions of *FaWRKY25* in strawberries’ resistance against *B. cinerea* and to clarify the functional mechanism of this gene and its interaction with JA during different fruit development stages, we investigated the transient overexpression and silencing of *FaWRKY25* in strawberries by using improved transient gene transformation technology. The modes of expression in the potential disease-resistant genes regulated by *FaWRKY25* and effects of external JA treatment to *FaWRKY25* expression were also analyzed.

## 2. Materials and Methods

### 2.1. Plant Materials

An octaploid strawberry cultivar, “Benihoppe” (*Fragaria* × *ananassa* Duch.), was employed. The experiment was conducted in an elevated strawberry planting base in a glass greenhouse in the agricultural expo park of Jiangsu Vocational College of Agriculture and Forestry from November 2018 to March 2020. No pesticides were used during the experiment. The experiment was performed at the daytime/nighttime greenhouse temperature of 20–23 °C/10–15 °C, with the strawberries growing in favorable conditions and without diseases or pests. The fruit development stages of the strawberries were divided into 7 periods: small green (SG; 15–20 days after flowering), medium green (MG; 20–25 days after flowering), big green (BG; 25–30 days after flowering), white (Wh; 30–35 days after flowering), turn red (Tu; 35–38 days after flowering), half red (HF; 38–40 days after flowering), and fully red (Re; 40–45 days after flowering) [44,45,46]. After the fruits were harvested, their surfaces were disinfected with 1.5% (*w/v*) sodium hypochlorite for 1 min, washed with distilled water 3 times, and air-dried in a laminar flow cabinet before being processed.

### 2.2. Inoculating B. cinerea in the Fruits That Received Different Treatments

The *B. cinerea* strains were acquired through separation from the fruits afflicted with *B. cinerea* gathered from the base of Jiangsu Agricultural Expo Garden by Jiangsu Horticultural Modern Engineering Center [47]. The separated strains were cultured in a culture medium with potato dextrose agar at 20 °C and a photoperiod of 16/8. Before the strains were inoculated in the strawberries, we cultured them in a medium with strawberry agar (500 g/L of ground strawberries and 1.5% of bacterial agar) to enhance their invasiveness [35]. Mycelia were scraped from the surface of the strains after 2 weeks of culturing, and conidia suspension was prepared in sterile distilled water with 0.03% (*v/v*) of Tween-80 at a concentration of 10^6^ spores per milliliter. A sterile filter paper with 3 layers was then used to remove the mycelia, and Neubauer cell chamber was employed to quantify the conidia concentration. Finally, the suspension with a *B. cinerea* concentration of 10^5^ spore/mL was prepared to infect the strawberries. Strawberries of even sizes, consistent levels of maturity, and normal shapes were harvested at different growth periods, with carpopodia of certain lengths preserved. They were then brought to the laboratory for *B. cinerea* inoculation. The fruits were treated with different concentrations of MeJA before being inoculated with *B. cinerea*, and the optimal MeJA concentration was determined according to the incidence of the fruits. The strawberries in the HF period were used for MeJA treatments, and fruits were inoculated with *B. cinerea* right after they were treated with different concentration of MeJA. The experiment was repeated 3 times, and 30 fruits were used in each batch for each treatment.

An inoculating needle was used to form 2 holes in the center (about 0.2 cm depth) of each strawberry fruit. The pathogen solution was filled into a wide mouth glass jar. Held at the carpopodia, the strawberries were immersed in the solution for 1 min before being placed in culture media for culturing at the appropriate humidity. The incidence of the fruits was followed up through photography and text recording.

### 2.3. Measuring JA Contents

Strawberries at different growth periods were harvested from the field. The achenes were rapidly removed from the fruits, which were then frozen using liquid nitrogen. The JA contents were measured using 200 mg of the evenly mixed homogenates of all fruits from each growth period; the experiment was performed according the method described by Kilam [48], with specific modification. The experiment was repeated 3 times, and 30 fruits were used in each batch for each developmental stage. 

Fruits at HF stage were treated with different concentrations of MeJA before being inoculated with *B. cinerea*, and the optimal MeJA concentration was determined according to the incidence of the fruits. The experiment was repeated 3 times, and 30 fruits were used in each batch for each treatment.

### 2.4. FaWRKY25 Gene Characteristics, Cloning, and Vector Construction

Clustal X 2.1 and ESprint 3.0 were implemented for the multiple alignment of amino acid sequences in the WRKY25 genes in the strawberries with the WRKY proteins of other plant species. The neighbor connection in MEGA X was applied to construct the phylogenetic tree.

Plant RNA extraction kits manufactured by Takara Company were employed to extract the total RNA of the strawberries, and complementary deoxyribonucleic acid was acquired through reverse transcription. According to the gene sequences recorded by the National Center of Biotechnology Information (accession number: XM_004294710.2), we applied DNAMAN to design the full-length primers of the target gene *FaWRKY25*. The cloned gene fragments had their sequences verified. *FaWRKY25* was constructed in the forward orientation on the Gateway expression vector pH7WG2D [47] to obtain the vector of *FaWRKY25* overexpression, 25-OE. To generate the *FaWRKY25* silent expression construct, we used vector pFGC5941 [47] to construct the hairpin structure of *FaWRKY25* (pFGC5941/*FaWRKY25*/RNAi (ribonucleic acid interference)). Then *FaWRKY25*/RNAi was PCR amplified using specify primers (Appendix A), which was then cloned to acquire the RNAi vector of *FaWRKY25* silencing, 25-RNAi. The empty vector pH7WG2D was employed as the control group. All the vectors were imported to *Agrobacterium tumefaciens* GV3101 through freeze–thawing for subsequent transient gene transformation in the strawberries. All the amplified sequences and specific primers used to construct the vectors are shown in Appendix A.

### 2.5. Transient Gene Transformation Methods

Transient gene transformation experiment was performed according the method described by Geng [44], with specific modifications. An inoculation loop was used to dip the agrobacterium solution carrying the genes, and then it was placed into a solid Luria–Bertan (LB) culture medium, sealed with a parafilm, wrapped in newspaper, placed upside down, and cultured in the dark for 2–3 days until a colony was formed. The colony was placed in a 50 mL Erlenmeyer flask with 10 mL of liquid LB. The flask was placed on a shaker and oscillated for a logarithmic period. A PCR test was run to ensure the bacterial solution was normal. Subsequently, 5 mL of the solution was extracted and transferred to a culture medium with 50 mL of A1, which was created by adding 1952 mg of MES and 3.924 mg of AS in each L of liquid LB. The solution was oscillated until the OD600 value was 1.0. The solution was centrifuged at low speed, resuspended in 50 mL of A2 (created by adding 1952 mg of MES and 39.24 mg of AS in a sterile MgCl_2_ solution), and oscillated at room temperature for 2 h before being injected into the strawberries. In vitro injection was performed on the strawberries during their Wh period. Through the use of a sterile syringe, we evenly injected 1 mL of the bacterial solution into the hollow strawberry fruit through the fruit stalk. The injection depth was approximately half of the longitudinal diameter of the fruit and the injection volume of *Agrobacterium* suspension was adjusted according to the size of the fruit to ensure the *Agrobacterium* suspension completely infected the fruit. Since the vectors used in the transient transformation study carried the e-GFP gene, the *FaWRKY25* gene expression was easy to be recorded through a fluorescence observation system, once successfully transformed.

### 2.6. Gene Expression Analysis

A Roche light cycler was applied in the real-time quantitative PCR system to measure the expression of all genes, and the internal reference gene was *FaActin*. The reaction system was 20 µL in volume and comprised 1 µL of complementary deoxyribonucleic acid, 10 µL of Til RNase Plus (2×), 1 µL of mixture of upstream and downstream primers, and 8 µL of ddH_2_O without nucleic acid contamination. Through the use of the 2^−∆∆CT^ method, we analyzed the relative gene expression data. The *FaWRKY25* expression during the green period was established as a control group to those of other growth periods and was defined as 1. For the gene expression after *B. cinerea* infection, the first day of infection was designated as the control group, with each gene expression defined as 1.

The following genes were selected for analysis in the fruits with *FaWRKY25* overexpression or silencing: *FaLOX*, *FaAOS*, *FaAOC*, *FaOPR2*, *FaOPR3*, *FaJAR1*, *FaJAR2*, *FaCOI1*, *FaMYC2*, *FaJAZ1*, *FaJAZ4*, *FaJAZ5*, *FaJAZ8*, *FaJAZ10*, and *FaJAZ12* for those related to the path of JA signal transduction; *FaWRKY1*, *FaWRKY2*, *FaWRKY11*, *FaWRKY33*, *FaWRKY40*, *FaWRKY57*, *FaWRKY70*, and *FaWRKY75* for those related to the WRKY family and the disease; and *FaBG2-1*, *FaBG2-2*, *FaBG2-3*, *FaPGIP1*, *FaPGIP2*, *FaCHI2-2*, and *FaCHI3-1* for those related to the structure of pathogen defense. The relative expression of these genes was depicted using a heat map. With the wild strawberries without bacterial injection as the control group, the relative gene expression in the empty vectors (EV-OE or EV-As), *FaWRKY25-*OE, and *FaWRKY25*-RNAi groups were calculated. The expression in the *FaWRKY25*-OE and *FaWRKY25*-RNAi groups was than divided by that in the empty vector group, and the results were used to create a heat map. All the gene information and primer sequences are shown in Appendix A.

### 2.7. Statistical Analysis

The experiment was conducted with a completely randomized design. Analysis of variance was performed to compare the samples’ incidence and the means of gene expression. The gene expression was normalized through logarithmic transformation to compensate for the abnormal distribution of data, with *p* ≤ 0.05 indicating statistical significance (least significant difference and Fisher’s method). SPSS 12 was used to complete the statistical analysis.

## 3. Results 

### 3.1. FaWRKY25 Protein Sequence Alignment and Phylogenic Tree Analysis

WRKY proteins are one of the largest TF families in plants and can be divided into three groups according to the number of WRKY domains and the type of zinc finger structure [49,50,51]. Group I includes two WRKY domains and one C2H2 zinc finger, group II includes one WRKY domain and one C2H2 zinc finger, and group III includes one WRKY domain and one C2HC zinc finger (C-X7-C-X23-H-X1-C). As shown in Figure 1A and Appendix A, *FaWRKY25* and *PtrWRKY25* were clustered more closely than *FaWRKY25* and *CpWRKY25*, indicating that the *FaWRKY25* in strawberries shares a close relationship with those in papayas and poplars and a distant relationship with those in *Camelina sativa* and *Zea mays*. The genetic codes of *FaWRKY25* encompass 627 amino acids, which include two typical WRKY domains and a C2H2 zinc finger with a structure consistent with that in the group I TF (Figure 1B and Appendix A). Accordingly, *FaWRKY25* belongs to the group I WRKY in strawberries.

### 3.2. Effect of JA Concentration on B. cinerea in Strawberries

As illustrated in Table 1, the effect of JA on *B. cinerea* in strawberries depends on its concentration. Although all strawberry samples with JA applied had their *B. cinerea* occurrence delayed compared with that in the control group (treated with clean water), a higher JA concentration did not necessarily lead to more effective *B. cinerea* containment. Within the range of 100–250 µM, a higher JA concentration delayed *B. cinerea* occurrence further; at 250 µM, the effectiveness of JA was the highest. However, a higher JA concentration beyond 250 µM led to earlier *B. cinerea* occurrence; observations revealed that increased JA concentrations led to fruit rot, which can render fruits vulnerable to *B. cinerea*. The severity of *B. cinerea* on the fruits according to the JA concentration was also examined (Table 2). The severity of the damage to the fruit tissues was divided into four levels: visible mild lesion (<10% fruit damage), moderate lesion (10–25% fruit damage), expanded lesion (25–50% fruit damage), and severe lesion (>50% fruit damage). The trend of the *B. cinerea* occurrence rate was consistent with the JA concentration—120 h after the *B. cinerea* infection, the fruits treated with 250 µM of JA exhibited only mild lesions (3.1% class I lesion). In the control group, 48 h after infection, 10% class I lesion and 10.2% class II lesion were detected; by the 120th hour after infection, 21.8% class IV lesion was identified, and most of the fruits were afflicted with *B. cinerea*. When the JA concentration was <250 µM, a higher concentration delayed and mitigated *B. cinerea* occurrence more effectively; however, when the concentration was >250 µM, the fruits became susceptible to the disease. At the 144th hour after *B. cinerea* infection, the fruits treated with 350 µM of JA exhibited 0.9% class IV lesion and 23.3% class I–III lesion. Accordingly, only within a certain range of concentration can JA effectively delay and mitigate the occurrence of *B. cinerea* on strawberries and increase the fruits’ resistance against *B. cinerea*. According to the results of this experiment, 250 µM is the optimal JA concentration, and thus this concentration was adopted in the following experiments.

### 3.3. Analysis of the FaWRKY25 Expression and JA Concentration in Strawberry Fruit alongside Maturation and in Response to B. cinerea

The JA content in strawberries changes significantly during fruit growth. From the SG to Re periods, the JA content rose first but dropped subsequently. During the BG period, the concentration was the highest; it decreased sharply starting from the Wh period and reached the lowest level in the Re period (Figure 2B). The level of *FaWRKY25* expression also varied considerably during the growth of strawberry fruits. As shown in Figure 2C, from the SG to Re periods, the *FaWRKY25* expression increased continually even though it decreased slightly in the HF period; starting from the Wh period, the expression increased rapidly and reached the highest point when the fruits were thoroughly red. 

To confirm the regulating effects of JA and *FaWRKY25* on strawberries’ *B. cinerea* resistance, we applied external JA and *B. cinerea* to the Re fruits in this study. The results revealed that *B. cinerea* occurred in the control group 120 h after infection; *B. cinerea* began to appear at the 48th hour in the *B. cinerea* infection group and grew increasingly severe over time, and at the 144th hour, more than 60% of the fruits were infected, and they rotted severely. The fruits that received JA treatment before *B. cinerea* infection remained healthy until the 144th hour, at which point visible *B. cinerea* began to appear on the surfaces of the fruits (Figure 2E). At the same time, changes in *FaWRKY25* expression occurred. In the control group, the *FaWRKY25* expression increased over time and peaked at the 144th hour; after the fruits were infected with *B. cinerea*, drastic changes occurred in *FaWRKY25* expression. Then, 48 h after infection, by which point visible *B. cinerea* had appeared, *FaWRKY25* expression increased substantially. As the infection grew increasingly severe, *FaWRKY25* expression rose sharply and reached twice the level of that in the control group at the 144th hour. In the fruits treated with external JA, which stayed healthy for 144 h, *FaWRKY25* expression was maintained at a low level throughout the period. These fruits became substantially resistant against *B. cinerea* and only exhibited mild visible *B. cinerea* 144 h after infection, and their *FaWRKY25* expression was significantly lower than that in the control and *B. cinerea* infection groups (Figure 2D). 

### 3.4. Regulatory Effect of Transient FaWRKY25 Expression in Strawberries on B. cinerea

As depicted in Figure 3A, the e-GFP started its faint expression 3 days after the agrobacterium injection; starting from the third day, the fruit started to turn red. Over time, the e-GFP expression increased gradually; by the fifth day, a minimum of 80% green fluorescent light was observed in most of the fruits. Through real-time fluorescence quantitative polymerase chain reaction (RT-qPCR), a significant change in *FaWRKY25* expression was detected 5 days after the agrobacterium injection. Compared with those in the fruits without injection (WT) and the empty vector (EV), the *FaWRKY25* transcription level in the fruits with agrobacterium injection increased substantially because of *FaWRKY25* overexpression. *FaWRKY25*-ribonucleic acid interference (-RNAi) caused the transcription level to decrease, and no significant differences were observed between the fruits without injection and the empty vector in terms of *FaWRKY25* expression (Figure 3C). Accordingly, transient gene transformation enabled transient *FaWRKY25* overexpression or silencing in strawberries, and agrobacteria did not cause a change in the *FaWRKY25* expression in the fruits. Therefore, a *B. cinerea* infection test was started 5 days after the agrobacterium injection into the Wh fruits.

The *B. cinerea* infection test was performed on the fruits with transient gene transformation, without injection, and with empty vectors injected (control group). The results revealed that the disease began to occur on the no injection group and control group on the fourth or fifth day after infection (DABI); the fruits with *FaWRKY25* overexpression had a significantly advanced time of disease occurrence and increased disease severity. On the sixth DABI, most of the fruits exhibited lesions, and over 50% had rotted severely (Figure 3E). Compared with the no injection group and control group, the fruits with *FaWRKY25* silencing had the time of disease occurrence significantly delayed and disease severity mitigated. By the sixth day of infection, only 70% of the fruits began to exhibit minor lesions, and almost no fruits displayed severe lesions (Figure 3B,E, Appendix A). Accordingly, the level of *FaWRKY25* expression significantly affected strawberries’ resistance against *B. cinerea*. *FaWRKY25* silencing delayed the occurrence of *B. cinerea* in strawberries and reduced the incidence of the disease; *FaWRKY25* overexpression accelerated the occurrence of the disease as well as its incidence (Appendix A). *FaWRKY25* played a key role in regulating strawberries’ resistance against *B. cinerea*. Notably, *FaWRKY25* also affected the JA content in strawberries; the JA content was significantly lowered in the *FaWRKY25*-OE fruits but significantly heightened in the *FaWRKY25*-RNAi fruits (Figure 3D). This indicated that JA stimulates *FaWRKY25* and that *FaWRKY25* can mediate JA biosynthesis and metabolism. 

### 3.5. Effect of FaWRKY25 on the Genes Related to B. cinerea Resistance

To discover the regulation mechanism of *FaWRKY25* on the *B. cinerea* resistance, we tested genes in JA synthesis and signaling pathway. In the present study, expression profiles of the genes of the key enzymes in JA biosynthesis, such as *FaLOX, FaAOS, FaAOC, FaOPR2*, and *FaOPR3*, were analyzed and coded. After the agrobacteria were injected into the fruits, *FaWRKY25*-OE resulted in an increase in *FaWRKY25* expression, whereas *FaWRKY25*-RNAi resulted in a decrease in *FaWRKY25* transcript level, indicating that *FaWRKY25* gene expression was successfully manipulated in the strawberry fruits by using transient overexpression (OE) and RNAi techniques (Figure 4). As the fruits matured, the *FaLOX, FaAOS, FaAOC, FaOPR2*, and *FaOPR3* expression decreased in the *FaWRKY25*-OE fruits, but the expression of the aforementioned genes exhibited an increase in the *FaWRKY25*-RNAi fruits, with the exception of *FaOPR2* (Figure 4). After the fruits were infected with *A. tumefaciens*, as time passed, the fruits with *FaWRKY25* overexpression exhibited symptoms of *B. cinerea* the earliest, which exacerbated rapidly; the JA content dropped significantly (Figure 3), and the *FaAOS*, *FaOPR2*, and *FaOPR3* expression decreased slightly (Figure 4). Furthermore, the manipulation of *FaWRKY25* expression in the fruit caused changes in the expression of some genes related to JA signaling pathway. In the *FaWRKY25*-OE fruits, the expression of *FaJAR1* and *FaJAR2* rose slightly, whereas the expression of *FaCOI1*, *FaMYC2,* and *FaJAZ8* showed a trend of decreasing first and then increasing. In the *FaWRKY25*-RNAi fruits, the expression of *FaCOI1, FaMYC2,* and *FaJAZ12* rose, whereas *FaJAZ4* expression dropped; the expression of *FaJAZ1* and *FaJAZ8* showed a trend of decreasing first and then increasing (Figure 4). Accordingly, *FaWRKY25* may have participated in JA biosynthesis and signaling, thereby influencing strawberries’ JA-mediated resistance against *B. cinerea*.

WRKY TFs feature numerous family members, and other unmentioned WRKY family members may play a role in regulating the JA-mediated resistance of strawberries against the pathogen. Therefore, this study investigated the expression of these other WRKY members, revealing that most of the WRKY TFs exhibited different degrees of change in expression. In the *FaWRKY25*-OE fruits, the expression of *FaWRKY1* and *FaWRKY70* increased, whereas the expression of *FaWRKY11, FaWRKY33, FaWRKY40*, and *FaWRKY75* decreased. In the *FaWRKY25*-RNAi fruits, the expression of *FaWRKY11, FaWRKY33,* and *FaWRKY75* increased, whereas the expression of *FaWRKY1* and *FaWRKY57* decreased (Figure 4). Additionally, there was an increase in some defense genes including *FaBG2-1*, *FaBG2-2*, *FaBG2-3*, *FaPGIP2*, *FaCHI2-2,* and *FaCHI3-1* detected in *FaWRKY25*-OE fruits; a decrease of *FaPGIP2*, *FaCHI2-2,* and *FaCHI3-1* expression was found in *FaWRKY25*-RNAi fruits; and the expression of *FaBG2-1* and *FaBG2-2* showed a trend of decreasing first and then increasing. 

## 4. Discussion

Many studies have shown that the WRKY family plays an important role in hormone regulation of disease resistance and defense signal transduction network. WRKY gene expression can be induced by pathogenic bacteria and hormone signal molecules such as jasmonic acid (JA) [52] For example, the upregulation of *AtWRKY70* in *Arabidopsis* can enhance the resistance to *B. cinerea*. *AtWRKY70* is a negative regulator of JA-induced gene, which can resist the infection of *B. cinerea* and coordinate the JA/ethylene of salicylate (ET) signaling pathway that plays a complex role in the cross-response process [27]; *AtWRKY18*, *AtWRKY40,* and *AtWRKY60* genes jointly regulate the resistance response of *Arabidopsis* to *B. cinerea* [28]. Earlier studies in *Arabidopsis* have shown that *AtWRKY33* is required for resistance to *B. cinerea,* implicating this factor to positively regulate JA- and ET-mediated defenses while negatively affecting SA-mediated signaling [22]. Later research showed that early JA signaling upon *B. cinerea* infection is independent of *AtWRKY33* function, while *AtWRKY33* acts as a negative regulator of SA signaling and thereby prevents subsequent SA-mediated suppression of the JA pathway [22,29,53]. These studies indicate that WRKY regulates the defense response of plants to *B. cinerea* in a variety of forms, such as single, cooperative, or complex. The diversity and complexity of WRKY regulation also reflected that the defense response of plants to *B. cinerea* is related to many factors. Transcription factors and hormones play an important role in plant regulation of *B. cinerea*.

Studies have shown that the WRKY gene plays an important role in strawberry disease resistance. According to previous research, we know that *FaWRKY1* is related to strawberry *C. acutatum* [34,35], and that *FvWRKY42* is a positive regulator of the resistance of wild strawberries against powdery mildew [37]. In 2016, 59 WRKY genes at different stages of fruit development were found in forest strawberry by transcriptome sequencing [31]. Subsequently, 62 WRKY genes responding to biotic and abiotic stresses were obtained from forest strawberry [54]. However, it is not known the role of WRKY in regulation of strawberry *B. cinerea*. As an important member of WRKY family, the function of *FaWRKY25* in strawberry resistance to *B. cinerea* and its relationship with the JA pathway are still unknown. To gain insight into the role of *FaWRKY25* in the strawberry fruit defense response to *B. cinerea*, we first confirmed the incidence of *B. cinerea* in strawberry fruits at different developmental stages. The results showed that green fruits were not susceptible to *B. cinerea,* while red fruits were susceptible to the *B. cinerea* (Appendix A). *FaWRKY25* gene expression increased rapidly with fruit ripening. However, the JA content of strawberry fruit decreased with fruit ripening. After jasmonic acid treatment, the expression of *FaWRKY25* gene decreased significantly, but the incidence of *B. cinerea* was delayed (Figure 1). On the basis of these results, we preliminarily infer that the basic expression of *FaWRKY25* in plant cells may be sufficient to regulate a set of specific *FaWRKY25* response defense genes to optimize the adaptability of plants. However, with the development of strawberry fruit, the expression of *FaWRKY25* gene increased gradually. The reason why fruit is prone to disease after ripening may be that the basic expression of *FaWRKY25* in fruit cells is too high with fruit ripening, which affects the defense system [3]. According to the relationship between JA content and *FaWRKY25* gene expression, we can also preliminarily conclude that *FaWRKY25* could regulate strawberry resistance to *B. cinerea*, and *FaWRKY25* negatively regulates JA.

To further analyze the function of the *FaWRKY25 g*ene in the disease process caused by *B. cinerea* in strawberry fruit, we accomplished overexpression and silencing of the gene in the fruit through the transient *A. tumefaciens*-mediated transformation method. Although the transient expression of *A. tumefaciens* was applied to strawberry fruit, the method has limitations because *A. tumefaciens* per se is a plant pathogen, and thus the interaction between the plant and pathogens is hindered [35,49,55,56]. Thus, when treating fruit, the empty vectors (the control) were arranged respectively for vectors with overexpression and silencing. The selected vectors carried enhanced green fluorescent protein (e-GFP) with extraordinarily strong expression. The strawberries with their genes successfully transformed emitted green light when excited by the light source, thereby facilitating subsequent examination [44]. Further analysis revealed that *FaWRKY25* overexpression significantly reduced strawberries’ resistance against the pathogen, whereas *FaWRKY25* silencing substantially increased their resistance against it. Moreover, *FaWRKY25* overexpression significantly inhibited JA content, whereas *FaWRKY25* silencing substantially increased it (Figure 3D). Furthermore, we speculated that *FaWRKY25* gene and JA signal pathway interact in the regulation of strawberry fruit disease resistance, JA and *FaWRKY25* can affect strawberry fruit disease resistance, and JA can regulate the expression of *FaWRKY25*. *FaWRKY25* plays a key role in regulating strawberries’ resistance against *B. cinerea* and may negatively regulate their JA-mediated resistance mechanism against the pathogen [29,30,31]. 

In order to reveal the molecular mechanism of *FaWRKY25* in resistance to *B. cinerea*, we detected the expression levels of some transcription factors and genes related to resistance. Phytohormone is known to exert a crucial effect in regulating plant immunity, with SA and JA being the main participants [51]. Research on *Arabidopsis* indicated that WRKY33, through activating the JA signaling pathway, regulated the resistance of the induction system triggered by *B. cinerea* [53]. To further analyze the molecular mechanism of this regulation, we tested the expression levels of the genes related to JA biosynthesis and metabolism as well as disease resistance, thereby clarifying the effect of *FaWRKY25* on strawberries’ resistance against *B. cinerea* under the regulation of JA and the genes that may play a role in regulation. JA biosynthesis is initiated with α-linolenic acid and is catalyzed with 13-LOX, AOS, AOC, and OPR3; finally, through three steps of β-oxidation, (3R,7S)-JA is formed [57,58]. The results showed that overexpression of *FaWRKY25* gene decreased the expression of *FaLOX*, *FaAOS*, *FaAOC*, *FaOPR2,* and *FaOPR3*; the content of JA in fruit was also decreased. Contrary, in the fruits with *FaWRKY25* silencing, the JA content increased and the expression of five genes increased significantly (Figure 3 and Figure 4). Our results supported the hypothesis that *FaWRKY25* can suppress the JA synthesis. Within the cells, JAR1, which is a member of the GH3 family, catalyzed the synthesis between JA and isoleucine to form JA-Ile, which is an active form of JA signal transduction in the cells. With JA-Ile, the JAZ protein synthesized with the SCFCOI1 E3 ligase and was degraded with the 26S proteasome, triggering the expression of genes responsive to JA. However, *JAR1* and *JAR2* are significantly reduced in the *FaWRKY25*-RNAi fruit, and the opposite occurs in *FaWRKY25*-OE fruit. Therefore, reduced production of JA-Ile is expected in *FaWRKY25*-RNAi fruit. Low JA-Ile level promotes JAZ expression and JAZ proteins to negatively regulate JA signaling pathway, including many TFs that positively regulate JA-responsive genes [59]. The COI1-coded F-box protein, which is a key substance in the ubiquitin ligase E3 in the SCF complex, specifically identifies JA-Ile [12,60]. MYC2, which is a key member of the bHLH family, is a core regulator of the paths of JA signals in a plant; JAZ is a negative regulator of the path of JA signal responses, and the complex model COI1-JAZs-MYC2 is currently recognized as a core regulatory model of the path of JA signal transduction [61,62]. Our results showed that *FaCOI1*, *FaMYC2*, *FaJAZ1*, and *FaJAZ8* expression rose significantly, whereas *FaJAZ4* expression dropped significantly in the *FaWRKY25*-RNAi fruits, (Figure 4). Accordingly, *FaWRKY25* may have participated in JA biosynthesis and signaling, thereby influencing strawberries’ JA-mediated resistance against *B. cinerea*.

We also found that when *FaWRKY25* gene was overexpressed or silenced, the expression levels of *FaWRKY1*, *FaWRKY2*, *FaWRKY11*, *FaWRKY33*, *FaWRKY40*, *FaWRKY57*, *FaWRKY70*, and *FaWRKY75* in WRKY transcription factor family were significantly regulated. However, it should be noted that the regulation of *FaWRKY 25* gene on these eight genes is not the same. Overexpression of *FaWRKY 25* gene positively regulates the expression of *FaWRKY1* and *FaWRKY 70*, and negatively regulates the expression of *FaWRKY2*, *FaWRKY11*, *FaWRKY33*, *FaWRKY40,* and *FaWRKY57.* In these genes, *FaWRKY33* and *FaWRKY70* genes were positively regulated by *FaWRKY25* gene, and their expression levels were significantly increased in comparison with the control. *WRKY33* and *WRKY70* are also important genes related to disease resistance reported in many studies [18,27,53]. This indicated that *FaWRKY25* and other WRKY members are jointly regulated by JA and play a crucial role in regulating strawberries’ resistance to *B. cinerea*. 

This result further confirmed that WRKY transcription factors play an important role in plant response to biotic and abiotic stresses as well as a variety of signaling pathways. Both JA and *FaWRKY*s can affect the disease resistance of strawberry fruit, and JA can regulate the expression of *FaWRKY*s. In other words, when *B. cinerea* was stimulated, the JA signal system in strawberry fruit would be regulated rapidly, which would increase/decrease the expression of downstream defense genes by regulating the expression of *FaWRKY*s, thus inhibiting/promoting the growth and proliferation of *B. cinerea*, while FaWRKYs affected strawberry disease resistance by regulating JA resistance signal in strawberry fruit, thus realizing strawberry resistance to *B. cinerea* wherein the mold signal was responded to.

*FaWRKY25* gene was overexpressed or silenced, and pathogenesis-related (PR) proteins β-1,3-glucanases (FaBG2-1, FaBG2-2, and FaBG2-3), chitinases (FaCHI2-2 and FaCHI3-1), and polygalacturonase-inhibiting proteins (FaPGIP1 and FaPGIP2) were analyzed. PGIP and chitinase play a defensive role in strawberry fruit. PR proteins also exert a critical effect in the course of disease and necrosis of plants [11,46]. Our results similarly indicated that the expression levels of most of these defense genes were regulated to diverse extents. The results demonstrated that plants could perform diverse defenses simultaneously. The *FaWRKY25* gene might enhance the defensive function of strawberry against *B. cinerea* through raising the level of *PR* and *PGIP* genes, thereby delaying the occurrence of *B. cinerea* in strawberry fruit.

In summary, in this paper, a functional characterization of the *FaWRKY25* gene was accomplished in strawberry fruit. We provide evidence of the relevance between *FaWRKY25* and strawberry fruit disease resistance against *B. cinerea*. In sum, we found that *FaWRKY25* acts as a negative regulator of strawberry fruit resistance to *B. cinerea*. 

## 5. Conclusions

The present study demonstrated that *FaWRKY25* plays an important role in regulating strawberries’ resistance against *B. cinerea*. Manipulating *FaWRKY25* expression modulated the strawberry fruits’ resistance against *B. cinerea*; the JA content in fruit was highly influenced by manipulating *FaWRKY25* expression as well, indicating that *FaWRKY25* may negatively regulate the JA-mediated resistance mechanism against *B. cinerea*. However, strawberry fruit resistance to *B. cinerea* is a complicated biological phenomenon, and further studies need to be done to identify the upstream and downstream signaling components of *FaWRKY25* in the stress response to *B. cinerea*.

## Figures and Tables

**Figure 1 genes-12-00056-f001:**
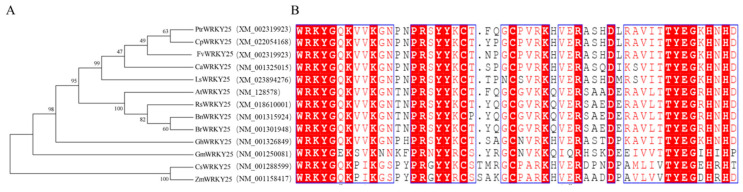
Phylogenetic analysis of WRKY25 protein. (**A**) Phylogenic analysis of WRKY25 homologs from various species. Species name abbreviations are follows: At, *Arabidopsis*; Br, *Brassica rapa*; Rs, *Raphanus sativus*; Bn, *Brassica napus*; Gh, *Gossypium hirsutum*; Gm, *Glycine max*; Cs, *Camelina sativa*; Zm, *Zea mays*; Ls, *Lactuca sativa*; Ca, *Capsicum annuum*; Cp, *Carica papaya*; Ptr, *Populus trichocarpa*. (**B**) Sequence alignment of the deduced amino acid sequences of WRKY25. The gene accession numbers are found in Appendix A.

**Figure 2 genes-12-00056-f002:**
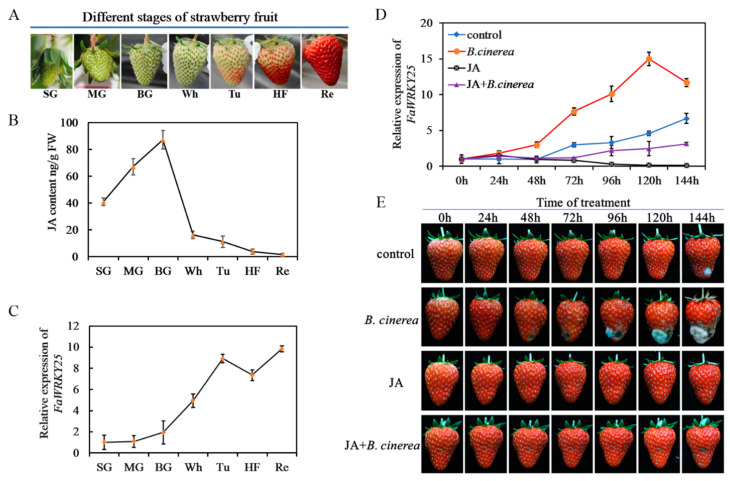
Variation of *FaWRKY25* gene expression and jasmonic acid (JA) content in strawberry fruits. (**A**) Phenotypes showing different developmental stages of strawberry fruit: SG (15–20 days after flowering), MG (20–25 days after flowering), BG (25–30 days after flowering), Wh (30–35 days after flowering), Tu (35–38 days after flowering), HF (38–40 days after flowering), and Re (40–45 days after flowering). (**B**) Changes in jasmonic acid levels throughout fruit development. (**C**) Changes in *FaWRKY25* gene expression levels throughout fruit development. (**D**) *FaWRKY25* relative expression in red fruit under different treatments. (**E**) Phenotypic effects of JA on red fruits after *Botrytis cinerea* inoculation. Control, fruits treated with sterile water; *B. cinerea*, fruits inoculated with *Botrytis cinerea*; JA, fruits treated with 250 µM JA; JA+*B. cinerea*, fruits inoculated with *Botrytis cinerea* after being treated with 250 µM JA. As for MeJA treatments, the fruits were inoculated with *B. cinerea* right after being treated with different concentrations of MeJA for 2 h. The experiment was repeated three times, and 30 fruits were used in each batch for each treatment.

**Figure 3 genes-12-00056-f003:**
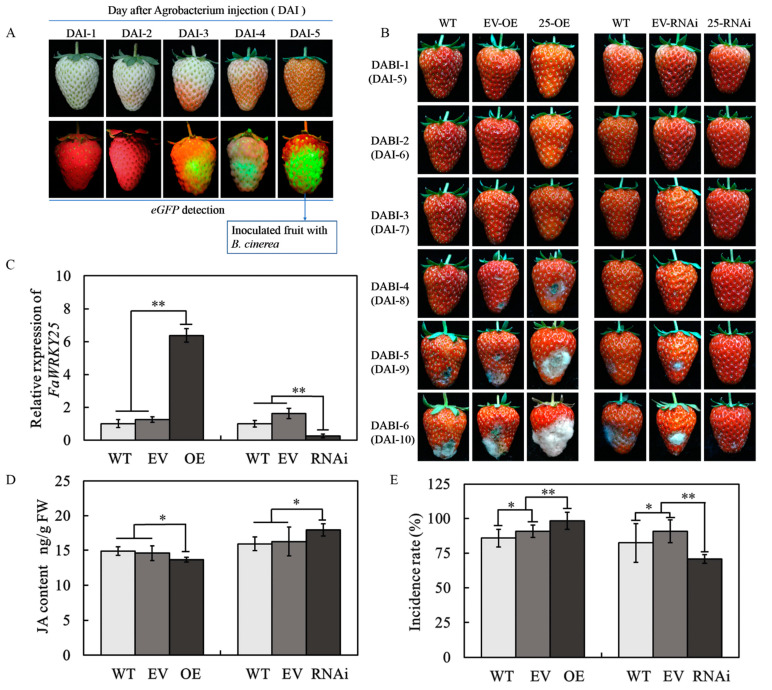
Effect of *FaWRKY25*-OE (-overexpression) and -RNAi (-ribonucleic acid interference) on the levels of *FaWRKY25* transcript and JA. (**A**) Expression of green fluorescent protein (eGFP) after *Agrobacterium tumefaciens* injection. (**B**) Phenotypes of *FaWRKY25*-OE and *FaWRKY25*-RNAi strawberry fruits after *Agrobacterium tumefaciens* injection and *Botrytis cinerea* inoculation. DABI denotes days after *Botrytis cinerea* inoculation, and DAI denotes days after *Agrobacterium tumefaciens* injection. (**C**) Effect of *FaWRKY25*-OE and -RNAi on *FaWRKY25* relative expression 5 days after *Agrobacterium tumefaciens* injection (DAI-5). (**D**) Effect of *FaWRKY25*-OE and -RNAi on jasmonic acid levels 5 days after *Agrobacterium tumefaciens* injection (DAI-5). (**E**) Comparison of the incidence rate in *FaWRKY25*-OE and *FaWRKY25*-RNAi strawberry fruits after *Botrytis cinerea* inoculation at DABI-6. Values are means ± SD of three biological replicates. Asterisks above the columns denote a significant difference at * *p* < 0.05 and ** *p* < 0.01 levels according to Student’s *t*-test.

**Figure 4 genes-12-00056-f004:**
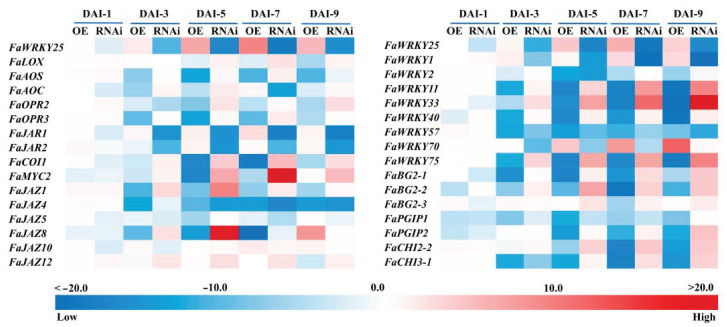
Effects of *FaWRKY25*-OE and -RNAi on transcription of resistance-related genes. OE and RNAi denote overexpression and silencing expression of the *FaWRKY25* gene. With the wild strawberries without bacterial injection as the control group, the relative gene expression in the empty vectors (EV-OE or EV-As), *FaWRKY25*-OE, and *FaWRKY25*-RNAi groups was calculated. The expression in the *FaWRKY25*-OE and *FaWRKY25*-RNAi groups was than divided by that in the empty vector group, and the results were used to create a heat map.

**Table 1 genes-12-00056-t001:** Incidence (%) of *Botrytis cinerea* in *Fragaria × ananassa* Duch. “Benihoppe” fruit after 3 days of treatment with different concentrations of MeJA.

Cultivars	MeJA Concentration (μM)	Treatment	Hours Post Inoculation						
			0	24	48	72	96	120	144
*Benihoppe*	0(CK,H_2_O)	+Bc	-	-	+	+	++	+++	++++
	100		-	-	+	+	++	+++	+++
	150		-	-	+	+	++	++	+++
	200		-	-	-	+	+	++	++
	250		-	-	-	-	-	+	+
	300		-	-	-	-	+	+	+
	350		-	-	-	-	+	+	++

Visual evaluation (-: absence; +, ++, +++, ++++: 0–25%, 25–50%, 50–75%, or 75–100% of fruit infected, respectively).

**Table 2 genes-12-00056-t002:** Comparison of the fruit tissue damage levels of strawberry after inoculation with *B. cinerea* in different treatments with different concentrations of jasmonic acid. Level 1 does not include any damage. The experiment was repeated three times, and 60 fruits were used in each batch for each treatment.

MeJA Concentration (μM)	Incidence (%) of Strawberry Fruit After Inoculation with *Botrytis cinerea*																			
	48 h				72 h				96 h				120 h				144 h			
	1	2	3	4	1	2	3	4	1	2	3	4	1	2	3	4	1	2	3	4
0(CK,H_2_O)	10.0 a	10.2	-	-	9.1 a	10.9 a	4.0 *	-	20.7 a	14.0 a	8.9 a	3.3	18.4 b	15.3 a	18.4 a	21.8 a	18.2 b	11.3 ab	26.2 a	36.7 a
100	6.2 b	-	-	-	7.8 b	8.4 a	0.7	-	15.8 b	12.9 a	5.3 b	3.1	22.0 a	12.7 b	15.8 ab	20.9 a	12.9 cd	13.3 a	24.2 a	24.9 b
150	3.3 c	-	-	-	5.8 c	3.6 b	-	-	11.3 c	12.2 a	2.7 c	-	15.1 c	12.9 b	13.8 b	2.9 b	13.1 cd	14.9 a	15.3 b	19.3 c
200	-	-	-	-	2.7 d	-	-	-	7.6 d	4.4 b	0.4 d	-	16.0 bc	9.8 c	4.0 c	-	24.7 a	9.1 cd	4.9 c	4.4 d
250	-	-	-	-	-	-	-	-	-	-	-	-	3.1 e	-	-	-	6.9 e	-	-	-
300	-	-	-	-	-	-	-	-	4.2 e	-	-	-	9.6 d	3.6 d	-	-	9.3 de	8.9 cd	0.7 d	-
350	-	-	-	-	-	-	-	-	6.4 de	-	-	-	10.2 d	4.0 d	-	-	14.7 bc	6.4 d	2.2 cd	0.9 d

Different small letters and asterisks above the columns denote a significant difference at * *p* < 0.05 in Student’s *t*-test.

## Data Availability

The research data are not available for public download to respect subjects’ privacy.

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
