# Peer review of "Strawberry FaWRKY25 Transcription Factor Negatively Regulated the Resistance of Strawberry Fruits to Botrytis cinerea"

_genes, 2020, doi:10.3390/genes12010056_

Round 1
Reviewer 1 Report
To unravel the function of genes encoding key transcription factors in strawberry defense is of great scientific and applied interest. In this study authors explore the putative role of strawberry FaWRKY25 gene on defense to B. cinerea. Their results support a negative correlation between the JA content and the expression of FaWRKY25 and support the idea that FaWRKY25 negatively regulate resistance to B. cinerea by suppressing many jasmonic acid pathway-dependent genes. Methodology is adequate but description must be improved. RESULTS section must be thoroughly rewritten and improved as there are some mistakes and confused things. The same apply to Discussion, which must be rewritten and improved, especially on gene expression analysis, which is poorly treated. The title should be reconsidered. Also, extensive editing of English language and style is required all through. Substantial changes and other comments to consider are included in the attached revised document.

Author Response
Dear Ms. Betty Guo Assistant Editor and Reviewers:
On behalf of my co-authors, thank you very much for your letter dated November 16. We were pleased to know that our work was rated as potentially acceptable for publication in Journal, subject to adequate revision. We appreciate the reviewers’ comments on our manuscript entitled “Strawberry FaWRKY25 transcription factor negatively regulated the resistance of strawberry fruits to Botrytis cinerea by suppressing the jasmonic acid pathway” (ID: genes-998361). Those comments are very helpful for revising and improving our paper. We have studied the comments carefully and have made correction which we hope meet with approval. Please find my itemized responses in below and my revisions in the re-submitted files.
Responds to the reviewers’ comments:
To Reviewer #1:
Point 1: “This is not entirely true as FaJAR1 and FaJAR2 are not suppressed!”
Response 1: Thanks for your kind advice. We have rewritten the title.
Point 2: “extensive editing of English language and style is required all through the article”
Response 3: We removed this sentence.
Point 4: “ARE”
Response 4: Revised accordingly.
Point 11: “To what extend? It seems to me that this scratching method does not provide homogeneity between fruits. Can you describe it carefully?”.
Response 11: Revised accordingly.
Point 12: ““Three groups of strawberries ….” . It is not clear to me what does it mean. As far as I understand, this is for the results shown in Figure 2A and 2B. But why “three groups”? Do you mean three repetitions of 10 fruit /repetition as you state later? Please clarify.”
Response 12: The experiment was repeated three times, and 30 fruits were used in each batch for each developmental stage. We have revised accordingly.
Point 19:“Please, give accession number for each gene and their corresponding Arabidopsis thaliana orthologs (in Table S3, it should be fine). Taking into account the current genome information available for strawberry, the expression of what specific gene alleles from these genes are you checking in the OV-FaWRKY25 and silenced-FaWRKY25 transient experiments? I mean, are the designed primers able to detect all the alleles for each gene or do they detect only specific alleles for each gene? Please, clarify this point and estate the correct accession number for genes.”
Response 19: Thank you for your advice. We have added relevant information in the Supplementary Table S3.
Point 20: “This is not clear”
Response 20: We have rewritten this part.
Response 21: Thank you for your advice. We have made a new figure and also attached a new figure in the supplementary data (Supplementary Figure S3).
Point 22: “Insert here accession number for each gene, including FaWRKY25”
Response 22: We have revised accordingly. The gene accession numbers are found in the Supplementary Table S3.
Response 23: We have revised accordingly.
Point 24: “If data in Table 2 are in % incidence and severity of the damage was divided into 4 levels as indicated, why in 0 M MeJA adding the data presented within the four levels does not give 100% at 96H, 120h or 144h DABI? To me, level 1 (<10%) includes no damage. If not, specified. The same happens for the other treatments. How many fruit (total fruit) was tested /treatment?”
Response 24: Thanks for your advice. Level 1 does not include no damage. The experiment was repeated three times, and 50 fruits were used in each batch for each treatment. We have revised accordingly.
Response 27: We have revised accordingly.
Point 29: “This is not clear to me as no photo is provided of what you describe!!!”
Response 29: Thanks for your advice. We removed the inappropriate description.
Response 40: Thanks for your advice. We removed the inappropriate description.
Response 43: We have revised accordingly.
Point 45: “HOWEVER, JAR1 AND JAR2 ARE SIGNIFICANTLY REDUCED IN FAWRKY25 SILENCED FRUIT and the opposite happens in OE-FaWRKY25 fruit. THEREFORE, REDUCED PRODUCTION OF JA-Ile IS EXPECTED in FaWRKY25 SILENCED. Low JA-Ile level promotes JAZ expression and JAZ proteins to negatively regulate JA signaling pathway, including many TFs which positively regulate JA-responsive genes (Gimenez-Ibanez et al., 2016),
Furthermore, in FaWRKY25 silenced fruit INCREASE OF FaJAZ genes is also being produced except JAZ4 and JAZ5 but reduced expression of FaBG2-1, FaBG2-2, FaBG2-3, FaPGIP1, FaPGIP2, FaCHI2-2 and FaCHI3-1 is not detected. PLEASE, COMMENT ON THAT”
Response 45: Thank you very much for your advice. We have revised accordingly.
Point 46: “This conclusion must be soften. There is no prove at all in this paper that other members of strawberry wrky family play critical role in strawberry defense against B. cinerea”.
Response 46: Thanks for your kind advice. We have revised accordingly.

Reviewer 2 Report
The abstract needs substantial improvement and requires a better structure. Please state the conclusion of the work in the abstract.
Line 6 “nearly none” meaning unclear please reword to improve English
Line 21 [increase] (Square bracket denote that text should be inserted in the manuscript)
Line 37 – I would say diploid Vesca is a model plant – octoploid strawberry cannot be considered a model plant – please specify.
Line 38 – the strawberry octoploid genome is certainly not small! This is factually incorrect
Line 43- it could be argued that oomycetes are the most threatening – given appropriate integrated pest management botrytis is more important as a post-harvest disease.
Line 48 – “can be lost by” poor English please reword
Line 61 to -> for
Line 69 in [the] plants
Line 71- what are “nutritional tissues “ this is an odd term
Line 81 first “of” to in
Line 85 Furthermore, [a] few
Line 92 odd wording please rephrase so meaning is clear: only provide a fundamental theoretical reference”
Line 98-100 meaning of sentence not clear please reword
Line 114 - FaWRKY25 [mediated] regulation
Methods – state numbers of strawberries assessed
Line 131 – specify host that isolate was taken from
Line 137: Procedure not clear: “A sterile filter paper was then used to remove the mycelia..”
Line 141: A more appropriate header would allude to the section containing the inoculation procedure
Line 151 achenes not seeds
Line 153: [the] experiment
Line 154: specific modification
Shouldn’t Table 1 be in the results?
Line 179 introductory sentence require
Line 184: use of word normal – this is meaningless to reader
Is morbidity the correct term here?
In figure 1 B sequence data does not appear to have been trimmed – trimming is required to represent the true phylogenetic relationship
The authors state:
The fruits with FaWRKY25 silencing did not differ significantly from the control group in lesion severity; this was because the agrobacteria affected the surface cells of the fruits less considerably than they affected the internal tissues.
How do the author support this statement? Can they be sure the RNAi was effective?
Discussion
Line 132: “ages does” missing words
Please comment on how a detached fruit assay may compare to infection of fruit on the plant.
Please comment on whether the pathogen may use interkingdom RNAi as part of the defence mechanism
Please common on the natural infection method in strawberry and how the infection method chosen influences your conclusions
Author Response
Responds to the reviewers’ comments:
To Reviewer #2:
Point 1: “The abstract needs substantial improvement and requires a better structure. Please state the conclusion of the work in the abstract.”
Response 1: Thanks for your kind advice. We have made some modification.
Point 4: “Line 37 – I would say diploid Vesca is a model plant – octoploid strawberry cannot be considered a model plant – please specify.”
Response 4: Thanks for your kind advice. We have revised accordingly.
Point 5: “Line 38 – the strawberry octoploid genome is certainly not small! This is factually incorrect”
Response 5: We have revised accordingly.
Point 6: “Line 43- it could be argued that oomycetes are the most threatening – given appropriate integrated pest management botrytis is more important as a post-harvest disease.”
Point 7: “Line 48 – “can be lost by” poor English please reword”.
Response 7: We have made modification.
Point 8: “Line 61 to -> for”
Point 9: “Line 69 in [the] plants”
Point 10: “Line 71- what are “nutritional tissues “ this is an odd term”
Response 10: We have made modification.
Point 11: “Line 81 first “of” to in”
Response 11: Revised accordingly.
Point 12: “Line 85 Furthermore, [a] few”
Response 12: Revised accordingly.
Point 13: “Line 92 odd wording please rephrase so meaning is clear: only provide a fundamental theoretical reference”
Response 13: We have removed this part according to the comment of Reviewer 1.
Point 14: “Line 98-100 meaning of sentence not clear please reword”
Response 14: We have removed this part according to the comment of Reviewer 1.
Point 15: “Line 114 - FaWRKY25 [mediated] regulation”
Response 15: Revised accordingly.
Point 16: “Methods – state numbers of strawberries assessed”
Response 16: We have added relevant information.
Point 17: “Line 131 – specify host that isolate was taken from”
Response 17: We have added relevant information.
Point 18: “Procedure not clear: “A sterile filter paper was then used to remove the mycelia.”
Response 18: We have made modification.
Point 19: “Line 141: A more appropriate header would allude to the section containing the inoculation procedure”
Response 19: Thanks for your kind advice. We have Removed this section title and link this paragraph to the previous one according to the comment of Reviewer 1.
Point 20: “Line 151 achenes not seeds”
Response 20: Revised accordingly.
Point 21: “Line 153: [the] experiment”
Response 21: Revised accordingly.
Point 22: “Line 154: specific modification”
Response 22: Revised accordingly.
Point 23: “Shouldn’t Table 1 be in the results?”
Response 23: Thanks for your advice. Revised accordingly.
Point 24: “Line 179 introductory sentence require”
Response 24: Revised accordingly.
Point 25: “Line 184: use of word normal – this is meaningless to reader”
Response 25: We have revised accordingly.
Point 26: “Is morbidity the correct term here?”
Response 26: Revised accordingly. We replace morbidity with incidence.
Point 27: “In figure 1 B sequence data does not appear to have been trimmed – trimming is required to represent the true phylogenetic relationship”
Response 27: Thank you for your advice. We have made a new figure in the supplementary data (Supplementary Figure S3).
Point 28: “The fruits with FaWRKY25 silencing did not differ significantly from the control group in lesion severity; this was because the agrobacteria affected the surface cells of the fruits less considerably than they affected the internal tissues. How do the author support this statement? Can they be sure the RNAi was effective?”
Response 28: Thanks for your kind advice. We removed the inappropriate description.
Point 29: “Line 132: “ages does” missing words”
Response 29: Revised accordingly.
Point 30: “Please comment on how a detached fruit assay may compare to infection of fruit on the plant.”, “Please comment on whether the pathogen may use interkingdom RNAi as part of the defence mechanism”, “Please common on the natural infection method in strawberry and how the infection method chosen influences your conclusions”.
Response 30: Thanks for your kind advice. In the present study, we focus on the analysis on the detached fruit, and further studies may be conducted on the fruit on the plant in the future. Furthermore, in the transient gene transformation experiment, even though the infection might have some effect on strawberry, we compare each treatment with the empty vector, and it will minimum the influence.

Reviewer 3 Report
The authors demonstrated that the transcription factor FaWRKY25 negatively regulates Jasmonic acid mediated resistance mechanism against Botrytis cinerea in strawberry fruits.
Over all experimental design and execution was oriented to address the problem.
The introduction and methods section need rephasing in some places for example
-line 48 should be more clear and specific
-line 71 In the nutritional tissue of c-doesn’t make sense-rephrase
-line 91-93 rephrase
-line 153 needs editorial correction
-line 174 Agrobacterium not Agrobacteria
-line 178 editorial correction
-line 196 rephrase
Table 2 legends are in different font sizes
In the methods section how many times the experiments were repeated not mentioned or how many fruits were tested for each treatment ( the average values mentioned is from how many units)
Some of the content in the discussion can be moved to results for example from line 134 to 150
is a reiteration of your results.
-line 136 also needs editorial correction
-Supplementary Table 1 was not cited any where in the text
Author Response
Responds to the reviewers’ comments:
To Reviewer #3:
Point 1: “line 48 should be more clear and specific”
Response 1: Revised accordingly.
Point 2: “line 71 In the nutritional tissue of c-doesn’t make sense-rephrase”
Response 2: Revised accordingly.
Point 3: “line 91-93 rephrase”
Response 3: We have removed this part according to the comment of Reviewer 1.
Point 4: “line 153 needs editorial correction”
Response 4: Revised accordingly.
Point 5: “line 174 Agrobacterium not Agrobacteria”
Response 5: Revised accordingly.
Point 6: “line 178 editorial correction”
Point 7: “line 196 rephrase”
Response 7: Revised accordingly.
Point 8: “Table 2 legends are in different font sizes”
Response 8: Revised accordingly.
Point 9: “In the methods section how many times the experiments were repeated not mentioned or how many fruits were tested for each treatment ( the average values mentioned is from how many units)”
Response 9: We have added relevant information.
Point 10: “Some of the content in the discussion can be moved to results for example from line 134 to 150 is a reiteration of your results. line 136 also needs editorial correction”
Response 10: Revised accordingly.
Point 11: “Supplementary Table 1 was not cited anywhere in the text”
Response 11: We have cited it in the revised manuscript.

Round 2
Reviewer 1 Report
This paper still need substantial corrections.
Authors should address all the questions properly and make comments on the contradictory results about gene expression of JA-responsive genes section.
Discusion needs further comment and reorganization.
Some errors still in this version.

Author Response
Dear Ms. Betty Guo Assistant Editor and Reviewer:
On behalf of my co-authors, thank you very much for your letter dated December 8.We appreciate the reviewers’ comments on our manuscript entitled “Strawberry FaWRKY25 transcription factor negatively regulated the resistance of strawberry fruits to Botrytis cinerea by suppressing the jasmonic acid pathway” (ID: genes-998361). Those comments are very helpful for revising and improving our paper. We have studied the comments carefully and have made correction which we hope meet with approval. Please find my itemized responses in below and my revisions in the re-submitted files.
Responds to the reviewer’s comments:
Response 1: Revised accordingly.
Point 2: “I would not say “crucial”. This is too strong”
Response 3: We removed this sentence.
Response 4: It is 2 weeks. Sorry for making mistake. We have revised it.
Point 11: “record”.
Response 11: Revised accordingly.
Response 12: We have revised accordingly.
Response 17: We have revised accordingly. We also rewritten the section of “3.5. Effect of FaWRKY25 on the Genes Related to B. cinerea Resistance”.
Point 19:“Which vector did you use?”
Response 19: Gateway expression vector pH7WG2D was used to obtain the vector of FaWRKY25 overexpression. One the other hand, vector pFGC5941 was used to construct the hairpin structure of FaWRKY25, which was then cloned to acquire the RNAi vector of FaWRKY25 silencing.
Point 20: “Discusion needs further comment and reorganization.”
Response 20: We have rewritten the discussion part according to the comments of reviewer.
Response 21: Thank you for your advice. We have revised accordingly.
Point 22: This paper still need substantial corrections.
Response 22: We have made substantial corrections to the paper, especially in the introduction and discussion sections, and have reorganized the references.
Point 23: Authors should address all the questions properly and make comments on the contradictory results about gene expression of JA-responsive genes section.
Response 23: We have rearranged the gene expression of JA-responsive section.
